# SARS-CoV-2 Myocarditis in a High School Athlete after COVID-19 and Its Implications for Clearance for Sports

**DOI:** 10.3390/children8060427

**Published:** 2021-05-21

**Authors:** Bibhuti B. Das

**Affiliations:** Department of Pediatrics, Heart Center, University of Mississippi Medical Center, 2500 North State Street, Jackson, MS 39216, USA; bdas@umc.edu; Tel.: +601-984-5250; Fax: +601-984-5283

**Keywords:** SARS-CoV-2, sports clearance after COVID-19, SARS-CoV-2 myocarditis, CMR imaging

## Abstract

This case report describes a high school athlete with palpitation, myalgia, fatigue, and dyspnea on exertion after SARS-CoV-2 infection with evidence of myocarditis by cardiac magnetic resonance (CMR), but echocardiography and troponin were normal. This case is unusual as the standard cardiac tests recommended by the American Heart Association for sports clearance, including ECG, echocardiography, and cardiac biomarkers, were normal. Still, she continued to be symptomatic after mild COVID-19. The CMR was performed to evaluate her unexplained palpitation and showed patchy myocardial edema two months after her initial SARS-CoV-2 infection. In this case, the diagnosis of myocardial involvement would be missed by normal echocardiograms and cardiac bio-markers without CMR. Because acute myocarditis is a risk factor for sudden death in competitive athletes, pediatric cardiologists should consider performing additional tests such as cardiac MRI in symptomatic COVID-19 patients, even if cardiac biomarkers and echocardiograms are normal.

## 1. Introduction

Severe acute respiratory syndrome coronavirus 2 (SARS-CoV-2) is the causative agent of coronavirus disease 2019 (COVID-19). COVID-19-associated myocarditis has been well described [1], but the occurrence of abnormal cardiac MRI (CMR) in athletes after COVID-19 is unknown. A prior study reported a modest (3%) prevalence of myocardial abnormalities among athletes experiencing mild illness or asymptomatic COVID-19 [2]. This case report describes a high school athlete with palpitation, fatigue, and exertional dyspnea after SARS-CoV-2 infection and was diagnosed to have myocardial edema by CMR. She had a normal echocardiogram and cardiac biomarkers. This particular presentation has not been described in an adolescent before and will add to the literature, bringing to light the significant variability of COVID-19 presentations. Also, myocarditis diagnosis in an adolescent with normal ECG (except sinus tachycardia) and echocardiogram has far-reaching implications, including returning to competitive sports and future cardiac complications.

## 2. Case Presentation

A 16 year old adolescent female presented to a pediatric cardiology clinic complaining of intense fatigue, myalgia, palpitations, and difficulty breathing for the last two months. She was positive for SARS-CoV-2 reverse transcriptase polymerase reaction (rt-PCR) on 15 July 2020, but she had no fever, cough, or difficulty in breathing at the time of the initial encounter and did not require hospitalization. Her rt-PCR test was negative for SARS-CoV-2 after two weeks of initial presentation. In the past, she was healthy and playing basketball for her school team. She had infectious mononucleosis two years previously, but otherwise had had no other significant medical illness in the past. She described her fatigue following COVID-19 as severe and precluding her from pursuing her usual activities. She was not able to participate in training and basketball practice. She complained of dyspnea, and her oxygen saturation dropped to the 80s with regular daily activities. Before the cardiology clinic visit, she had two visits to her primary care physician and one local ER visit between July and September last year. All her previous laboratory tests, including C-reactive protein (CRP), erythrocyte sedimentation rate (ESR), ferritin, d-dimer, troponin, B-type natriuretic peptide (BNP), troponin, complete blood count (CBC), and complete basic panel (renal and liver functions) were normal. A pulmonologist and rheumatologist had also seen her and could not find any reason for her fatigue. Her rheumatoid factor, thyroid function, vitamin D level, and anti-nuclear antibody screen were negative. A chest CT was reported as normal in September 2020. No cause was found for her intense fatigue, joint pain, and myalgia.

Upon arrival at the cardiology outpatient clinic on 20 September 2020, physical examination revealed she had resting tachycardia (heart rate 92/min), respiratory rate 24/min, blood pressure 106/65 mmHg, and oxygen saturation by pulse oximetry 97%. A six-minute walk test (6-MWT) showed she had decreased endurance (total distance was 380 m (normal 618 ± 79 m)), heart rate increased to 136/min, and her oxygen saturation dropped to 82%. Her ECG in the clinic showed a sinus rhythm with sinus tachycardia (Figure 1).

She had an echocardiogram in the clinic, which showed normal function and normal coronary anatomy. She underwent outpatient CMR imaging that showed patchy myocardial edema without any fibrosis or scarring (Figure 2A–F). In this case, a short tau inversion recovery sequence was used for T1 and T2 mapping, performed at the mid-cavity short-axis slice before and at 15 min after contrast administration according to our institutional protocol. The myocardial perfusion or regional wall motion was normal at rest. There was patchy myocardial edema on the short tau inversion recovery sequence, more pronounced along the anterior and anteroseptal apical left ventricular wall (Figure 2A). The average myocardial T2 relaxation time was >70 ms (normal, 44–49 ms) by quantitative T2 mapping, which suggested myocardial edema in the anterior and anteroseptal regions [3]. The T2 values of the myocardium were compared to that of the pectoralis muscle, and the relative myocardial signal intensity was calculated to be >2.4. Consistent with previous studies, this patient had myocardial edema, suggestive of acute mild myocarditis [4,5]. As shown in Figure 2B, the T1 mapping calculated average extracellular volume of fluid using a hematocrit of 40 was 25.1% (normal, 25.3 ± 3.5%). Figure 2C shows the global myocardial T1 relaxation time, averaged 1000 ± 30 ms, suggesting no evidence of diffuse fibrosis (normal 1250 ± 50 ms, according to our institutional protocol). Figure 2D shows the normal function of both ventricles and pulmonary arteries and lungs were within the standard limit. Figure 2E,F shows no early or late gadolinium enhancement to suggest the absence of fibrosis or scarring. In this patient, there was an increase in T2 values of myocardial relaxation time, though in patchy areas, suggestive of myocardial edema and commonly seen in children with myocarditis [6]. This finding of patchy edema can be missed by conventional magnetic resonance imaging [7]. In a recent expert panel, an update of the Lake Louise criteria for diagnosing myocarditis was proposed [3]. However, the accuracy of Lake Louise criteria for acute myocarditis is unsatisfactory and is more beneficial for chronic myocarditis rather than acute mild myocarditis [8]. The presence of myocardial edema is helpful to diagnose early acute mild myocarditis in adults [9,10]. There is an urgent need for uniform CMR guidelines for evaluating cardiac involvement in children after they have contracted COVID-19.

The patient was admitted to the hospital following the CMR for observation and monitoring. Her troponin was <0.01 ng/mL (N, <0.03), CRP <0.5 mg/dL (N, <1 mg/dL), ESR 12 mm/hr (N, 0–20 mm/hr), BNP 53 pg/mL (N, <100 pg/mL), ferritin 8 ng/mL (N, 10–70 ng/mL), D-dimer 0.61 mcg/mL (N, <0.4 mcg/mL), procalcitonin 0.3 ng/mL (N, 0.1–0.49 ng/mL), fibrinogen 374 mg/dL (N, 220–440 mg/dL), and CBC and metabolic panel, renal, and hepatic function were within the normal limit. Her repeat rt-PCR for SARS-CoV-2 was again negative at two months from the initial encounter. No SARS-CoV-2 antibody testing was done. She underwent pulmonary function tests (PFT), which showed normal functional vital capacity (84%), forced expiratory volume in one second (82%), ratio of forced expiratory capacity volume in one second and forced vital capacity (76%), and vital capacity (3.6 L), but a slightly decreased mid-expiratory flow rate (70%). She received a bronchodilator and steroid inhaler challenge but there was no significant change in PFT. No arrhythmia was noted during hospitalization and by Holter monitor. The next day, she was discharged home with symptomatic therapy and reassurance that her symptoms would get better over time. It was recommended that she not participate in competitive sports until a repeat cardiac MRI in 6 months.

## 3. Discussion

A characteristic of COVID-19 is the broad latitude of associated symptoms. One reason for this may be the varying immune response between patients, with mild cases assumed to have an effective response by the host. It has been established that COVID-19 can cause significant cardiovascular complications, including myocarditis, acute coronary syndrome, thromboembolism, and arrhythmias [11]. Puntmann and colleagues analyzed the CMR findings of SARS-CoV-2 in adults who experienced minimal COVID-19 symptoms on initial presentation and found 78% had demonstrable cardiac involvement suggestive of myocarditis 71 days after confirmed COVID-19 diagnosis [12]. Many of these patients are either asymptomatic or reported atypical chest pain, difficulty breathing, and palpitations but had abnormal CMR, including edema and/or fibrosis during convalescence. Of the CMR parameters studied, the authors reported that native T1 (indicating diffuse myocardial fibrosis) was elevated in 71% of patients, and 60% had elevated T2, a more specific marker for high water content and myocardial edema/inflammation when compared to healthy controls without COVID-19. Although the long-term clinical significance of the myocardial edema in CMR during COVID-19 recovery is unknown, CMR evidence of myocardial edema may carry prognostic value [13].

Figure 3 illustrates putative mechanisms for the involvement of the heart and cardiovascular system. Cardiac complications due to COVID-19 can occur either directly due to inflammation of the myocardium (myocarditis) or indirectly by one or a combination of mechanisms such as angiotensin-converting enzyme 2 (ACE2)-mediated loss of protective cardiovascular effects on target organs, hypoxia-induced excessive intracellular calcium leading to cardiac myocyte apoptosis, microvascular disease, and cytokine storm [14].

Tavazzi et al. reported one case demonstrating low-grade myocardial inflammation in the absence of myocardial necrosis [15]. The authors of this case did not observe any viral genome in myocytes; however, viral particles consistent with COVID-19 were found in macrophages in the interstitial tissue adjacent to cardiac myocytes, and therefore one cannot infer if SARS-CoV-2 is cardiotropic. Recently, Lindner et al. published the results of 39 autopsy cases from Germany, ages 78 to 89 years, who had tested positive for COVID-19. The authors found that 16 of the patients had the virus in their myocardial tissue samples but did not show signs of inflammation or myocarditis [16]. There is a possibility that ongoing inflammation may cause chronic myocarditis, and the actual incidence is unknown. An important distinction should be made between symptoms due to persistent chronic inflammation (convalescent phase) and organ damage sequelae.

The presence of cardiac abnormalities during COVID-19 convalescence in athletes is much less than in adults, as shown in the study by Putmann et al. (78%) [12] and Huang et al. (58%) [17]. In athletes, the utility of CMR is questionable as the prevalence of myocarditis is low (1.4%) [18]. In another study of 26 consecutive elite athletes who had asymptomatic to mild COVID-19 with no ECG abnormalities and normal troponins 1–2 months after diagnosis, 19% of them had some abnormalities in CMR but no definite acute myocarditis [19]. Although myocarditis is a significant cause of sudden cardiac death during exercise, there is no widely accepted definition of what constitutes clinically relevant myocardial injury secondary to COVID-19 among athletes. Simultaneously, there is an absence of a similar degree or completely benign postinfectious CMR findings with other respiratory viruses for comparison.

American Heart Association guidelines from 2015 recommend CMR in the setting of elevated troponin, arrhythmia, abnormal ECG, or echocardiogram as one of the screening tools to disqualify athletes from participating in competitive sports [20]. During the COVID-19 pandemic, increasing use of CMR may lead to an increase in false-positive rates. A special expert communication on return to play among high-school athletes aged >15 years after contracting COVID-19 recommended CMR only if clinical evaluation with ECG, echocardiogram, and troponin levels are abnormal, similar to AHA guidelines [21]. In summary, routine CMR as a screening tool for myocarditis in competitive student-athletes returning to training after recovering from COVID-19 is unnecessary. A recent misanalysis of the PubMed database through 2020 reported that since myocarditis can present with various symptoms and be asymptomatic, physicians need to have a high index of suspicion for its diagnosis. If there is any doubt regarding whether the myocardium is affected, physicians should obtain further testing, including CMR and an exercise test, as needed [22].

The interim statement from the American Academy of Pediatrics (AAP) for sports clearance after COVID-19 recommends that “all children and adolescents with exposure to SARS-CoV-2, regardless of symptoms, require a minimum 14-day resting period and must be asymptomatic for >14 days before returning to exercise and competition. Because of the limited information on COVID-19 and exercise, the AAP strongly encourages that all pediatric patients with COVID-19 be cleared for sports participation by their primary care physician.” [23]. The focus of sports participation screening should be on cardiac symptoms, including but not limited to chest pain, shortness of breath, fatigue, palpitations, or syncope. If the primary care physician is concerned about the patient’s cardiac involvement or readiness for sports participation, they should not hesitate to consult or refer to pediatric cardiologists. While multiple prospective studies are ongoing, there is an urgent need to develop uniform CMR criteria for diagnosing and managing SARS-CoV-2 myocarditis in children.

## 4. Conclusions

In our case, the patient was symptomatic, and her symptoms alluded to the post-COVID syndrome. However, she also had decreased exercise tolerance in 6-MWT and baseline tachycardia, for which CMR was performed. The CMR findings of patchy myocardial edema suggest acute mild myocarditis. This case’s implication is far-reaching for high-school athletes who participate in sports after COVID-19. The critical point for the physician who clears athletes for sports participation is to be vigilant for evidence of myocarditis. The present case poses a challenge as her echocardiogram and cardiac biomarkers were normal, but she was symptomatic. All symptomatic patients should undergo additional cardiac testing, including CMR, as deemed necessary.

## Figures and Tables

**Figure 1 children-08-00427-f001:**
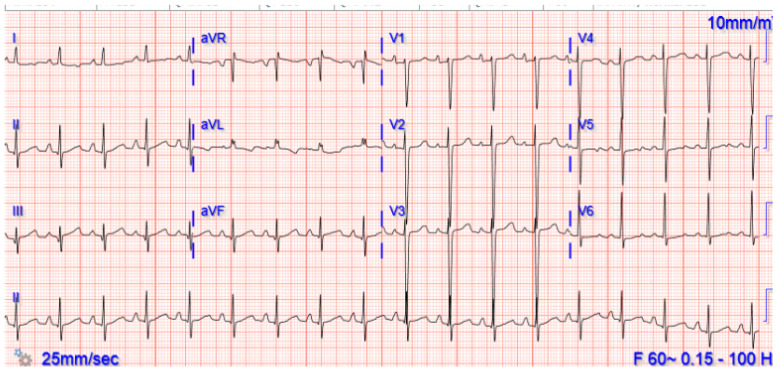
Sinus rhythm with sinus tachycardia.

**Figure 2 children-08-00427-f002:**
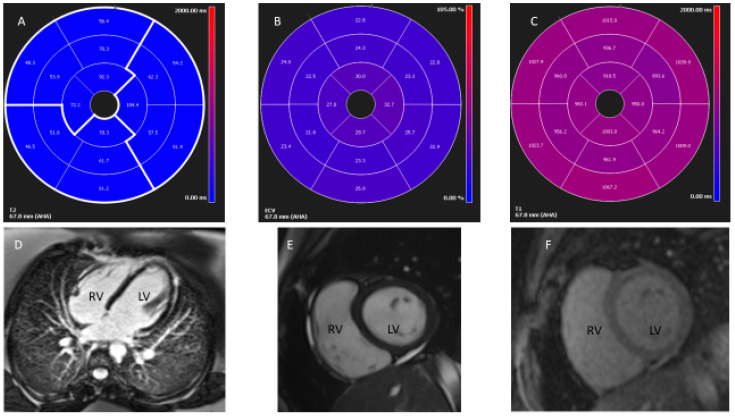
Cardiac magnetic resonance imaging showing the diagnosis of myocarditis. Cardiac magnetic resonance imaging: (**A**) T2 mapping showed elevated myocardial relaxation time consistent with patchy edema; values for the T2 ratio compared to skeletal muscle >2.4; (**B**) T1 mapping showed normal extracellular fluid volume; (**C**) T1 mapping showed no myocardial fibrosis; (**D**) Normal size left and right ventricles, normal lung parenchyma and branch pulmonary arteries; (**E**) absence of early gadolinium enhancement; and (**F**) absence of delayed myocardial enhancement.

**Figure 3 children-08-00427-f003:**
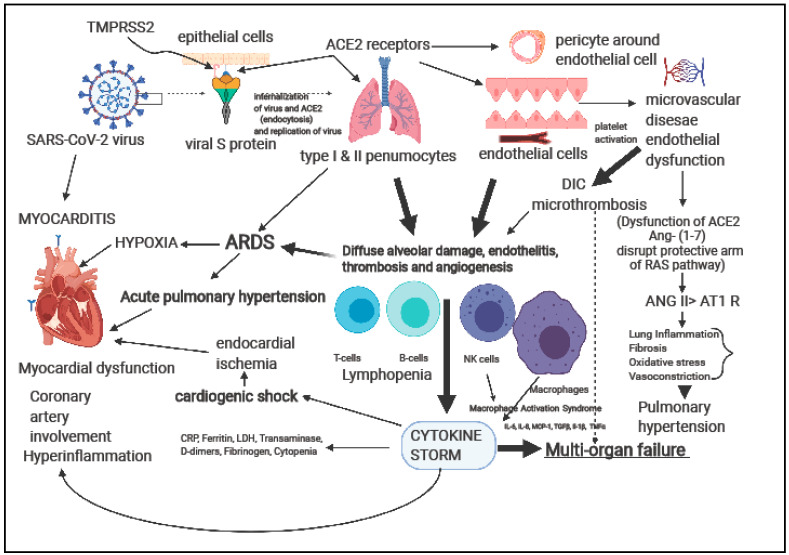
Several putative mechanisms of myocardial injury in early and late-COVID-19 illness. (Ang: Angiotensin; AT1 R: Angiotensin 1 receptor; ACE2: Angiotensin-Converting Enzyme; ARDS: acute respiratory disease syndrome; CRP: C-reactive protein; DIC: Disseminated intravascular coagulation; IL: Interleukin; LDH: lactate dehydrogenase; NK cells: natural killer cells; RAS: Renin-Angiotensin-System; TMPRSS2: transmembrane serine protease 2) (Created with biorender.com).

## Data Availability

No data available.

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
