# Peer review of "SARS-CoV-2 Myocarditis in a High School Athlete after COVID-19 and Its Implications for Clearance for Sports"

_children, 2021, doi:10.3390/children8060427_

Round 1
Reviewer 1 Report
Summary: case presentation of a child with persistent cardiac symptoms post-COVID infection with evidence of myocardial edema cardiac MRI.
Overall, I think it is important to report interesting cases of post-COVID cardiac involvement but I have a few major concerns about this report.
First, this is not a clear diagnosis of myocarditis and should not be stated as such. This can be called myocardial edema. As mentioned, the Lake Louise Criteria was created in the setting of clinically suspected myocarditis, and should include at least 2 criteria for high specificity of diagnosis. The patient presented has no clinical features of myocarditis (ECG, troponin, echo). It is a stretch to say this classifies as myocarditis.
Secondly, there are many missing features in the case presentation: timeline of COVID symptoms, timing of initial studies and results, timing of cardiology clinic visit and MRI, COVID antibody test, Chest x-ray, objective numbers for 60min walk test and PFTs.
Third, the manuscript was not well written with multiple grammatical errors and careless typos.
Lastly, I do not see the reason for including figure 2 and figure 3 in this case study. This case report is not meant to include proposed mechanisms of cardiac-injury or to propose a guideline for sports clearance. It is appropriate to mention those things and site prior published work, but should not be included in this report.
It seems the goal of this paper should be to present the case as is and then question whether there is any implication of detecting myocardial edema on cardiac MRI in post-COVID patients. Does this truly mean there is a diagnosis of myocarditis. Is edema itself linked to adverse outcomes? Is edema also present on other MRI after other viral infections. These would be more interesting questions to address.
Author Response
Summary: case presentation of a child with persistent cardiac symptoms post-COVID infection with evidence of myocardial edema cardiac MRI.
Overall, I think it is important to report interesting cases of post-COVID cardiac involvement but I have a few major concerns about this report.
First, this is not a clear diagnosis of myocarditis and should not be stated as such. This can be called myocardial edema. As mentioned, the Lake Louise Criteria was created in the setting of clinically suspected myocarditis, and should include at least 2 criteria for high specificity of diagnosis. The patient presented has no clinical features of myocarditis (ECG, troponin, echo). It is a stretch to say this classifies as myocarditis.
- Respectfully, disagree with the esteemed Reviewer. The lake Louise Criteria as in reference #4: states that edema suggests acute myocarditis, especially if the ratio of T2 value compared to skeletal muscle is >1.9: we added our value >2.4 (Additional Ref to estimate ratio of T2 compared to skeletal muscle (adjacent pectoralis muscle): Abdel-Aty H, Boye P, Zagrosek A, Wassmuth R, Kumar A, Messroghli D, Bock P, Dietz R, Friedrich MG, Schulz-Menger J. Diagnostic performance of cardiovascular magnetic resonance in patients with suspected acute myocarditis: comparison of different approaches. J Am Coll Cardiol. 2005;45:1815–22. [PubMed] [Google Scholar])
- Edema in the absence of fibrosis suggests reversible injury.
- Edema in the presence of early gadolinium enhancement is indicative of hyperemia and acute myocarditis.
Secondly, there are many missing features in the case presentation: timeline of COVID symptoms, timing of initial studies and results, timing of cardiology clinic visit and MRI, COVID antibody test, Chest x-ray, objective numbers for 60min walk test and PFTs.
- Thank you. I have summarized the timeline of symptoms and laboratory tests for the case report. COVID antibody test is not available. CXR and chest CT were normal. If the Reviewer feels those will add to the case report, happy to provide them.
Third, the manuscript was not well written with multiple grammatical errors and careless typos.
- Thank you for pointing it out; please let me know if there are still issues.
Lastly, I do not see the reason for including figure 2 and figure 3 in this case study. This case report is not meant to include proposed mechanisms of cardiac-injury or to propose a guideline for sports clearance. It is appropriate to mention those things and site prior published work, but should not be included in this report.
- I agree with the Reviewer and removed Figure-3 for sports clearance as there are now many available from national data and guidelines. I strongly feel the pathogenesis of myocarditis is essential for readers to understand, and Figure-2 is unchanged (revised manuscript, Figure 3).
It seems the goal of this paper should be to present the case as is and then question whether there is any implication of detecting myocardial edema on cardiac MRI in post-COVID patients. Does this truly mean there is a diagnosis of myocarditis. Is edema itself linked to adverse outcomes? Is edema also present on other MRI after other viral infections. These would be more interesting questions to address.
- Thank you for this critical point. I addressed the issue of CMT findings with other viral illnesses. There is an absence of a similar degree of CMR findings or completely benign postinfectious CMR findings with other respiratory viruses for comparison. This has been added to the revised manuscript. In HIV patients, similar sub-clinical CMR findings have been reported by CMR, but they have low-grade viremia.
Reviewer 2 Report
The author presents an interesting case of myocarditis in a young girl months after the COVID-19 infection.
The paper should be improved in many aspects starting from the abstract (too short), English language (and typos), more data in the Introduction, the case presentation (better to have a clear timeline - just one data 15 July and no other calendar data presented), and Discussion.
The management of the case and the follow-up should be presented in the Case presentation and then, if needed, be discussed in Discussions. The cardiac MRI aspects should not be repeated in the Discussion again after the figure 1 presentation inside the case report. It seems that this is not related to the case but only a theoretical discussion.
There is a discussion related to clearance for sport, but without any reference to the present case. She was involved in a professional sport before COVID-19?
I would clearly try to present one last paragraph as Conclusions.
The author should revise the use of abbreviated words in the paper.
The keywords should not have twice SARS-CoV2 and COVID-19.
References 3 and 5 should include 6 authors et al. if more than 6.
Author Response
Reviewer-2:
The author presents an interesting case of myocarditis in a young girl months after the COVID-19 infection.
The paper should be improved in many aspects starting from the abstract (too short), English language (and typos), more data in the Introduction, the case presentation (better to have a clear timeline - just one data July 15th and no other calendar data presented), and Discussion.
- Thank you. The revised manuscript is organized as Introduction, case report, Discussion, and Conclusions. I added the timeline from July until I saw the patient on September 20th.
The management of the case and the follow-up should be presented in the Case presentation and then, if needed, be discussed in Discussions. The cardiac MRI aspects should not be repeated in the Discussion again after the figure 1 presentation inside the case report. It seems that this is not related to the case but only a theoretical discussion.
- Thank you. The case report has been reorganized, and Figure-2 (previous Figure-1) is inside the case report.
There is a discussion related to clearance for sport, but without any reference to the present case. Was she involved in a professional sport before COVID-19?
- Thank you. Figure-3 is removed. The Discussion is relevant for sports participation as the patient is a high school athlete and discussed the current data from the literature in the discussion section.
I would clearly try to present one last paragraph as Conclusions.
- Thank you. The conclusion is added.
The author should revise the use of abbreviated words in the paper.
- Thank you. Most abbreviations have been removed.
The keywords should not have twice SARS-CoV2 and COVID-19.
- (SARS-CoV-2) is the causative agent of coronavirus disease 2019 (COVID-19). Most people are making this mistake.
References 3 and 5 should include 6 authors et al. if more than 6.
- All references with more than 6 authors have been added.
Reviewer 3 Report
I read this case report with great interest. To my opinion it clearly adds to the constantly growing knowledge on Cars-CoV-2 infection related conditions.
I only have 1 minor comment:
In the introduction the numbers of people affected by COVID worldwide should be corrected since currently they rose to quite some extent.
Author Response
I read this case report with great interest. To my opinion it clearly adds to the constantly growing knowledge on Cars-CoV-2 infection related conditions.
I only have 1 minor comment:
In the Introduction, the numbers of people affected by COVID worldwide should be corrected since currently, they rose to quite some extent.
- Thank you. The revised manuscript has removed this line as this is a moving target as there is a recent surge again with variants of SARS-CoV-2.
Reviewer 4 Report
I read with interest the article describing a case report of myocarditis occurring after COVID infection. The case need further revision before acceptance.
It’s nuclear to me if the author is describing isolated myocarditis that occurred after COVID or a case of MIS-C. The clinical presentation lacked sufficient information to assess if the patient met the CDC criteria fir MISC but the discussion start with discussing MIS-C.
Please provide image of ECG for readers to decide if it was normal.
The proposed return to sport algorithm would have missed this case as the patient was not admitted and MISC was not really present Ed in the case description. That being said, the return to sport part is out of the discussion of one case report.
Author Response
I read with interest the article describing a case report of myocarditis occurring after COVID infection. The case needs further revision before acceptance.
It's nuclear to me if the author is describing isolated myocarditis that occurred after COVID or a case of MIS-C. The clinical presentation lacked sufficient information to assess if the patient met the CDC criteria for MISC but the Discussion start with discussing MIS-C.
- Thank you. I revised the manuscript. This case had no fever and did not have MIS-C. This is a case of post-COVID myocarditis with unusual presentation.
Please provide image of ECG for readers to decide if it was normal.
- Thank you. ECG is added as Figure-1
The proposed return to sport algorithm would have missed this case as the patient was not admitted and MISC was not really present Ed in the case description. That being said, the return to sport part is out of the Discussion of one case report.
- Thank you. This young lady is a high school basket player. The case report is revised, and the title has also been revised.
Round 2
Reviewer 1 Report
I still have a lot of concerns about this paper. There continues to be very poor grammar, typos throughout, and a lack of data (and understanding of prior research) to support some of the claims in the paper. Overall, it is difficult to follow. While the case is interesting, the author is unable to convey the implications of the case and how it compares to other previous reports.
I have some line by line edits, but after writing out over 2 pages of suggested edits in a word document, I have found that there is too much.
Author Response
Thank you for continued review and positive feedback to improve the case report.
1. We added all previous case reports of myocarditis as reference 1.
(Das BB, Tejtel SKS, Deshpande S, Shekerdemian LS. Cardiac presentation of coronavirus disease 2019 (COVID-19) in adults and children. Texas Heart Inst J 2021;48(2):e20-7395)
2. We revised the paper with unequivocal literature support for the diagnosis of myocarditis by CMR in this case and its implications.
3. If the reviewer kindly specify all the concerns and mistakes: I am happy to clarify and correct the spelling and grammar mistakes. The paper was reviewed by a native English speaking professional and is approved.
Reviewer 2 Report
The manuscript was improved extensively based on some of the comments, and it seems better now. In the present form, it is clearer the clinical evolution and the discussion of all implications from an athlete's point of view.
Still, authors should correct the number of figure 2 inside the moved text (still, there are references to figure 1 in lines128-135)
For keywords, I proposed not to use COVID-19 also in the third word and SARS-CoV-2 in the fourth as already the first two words describe the virus and the disease. That was the reason for my comment.
Author Response
The manuscript was improved extensively based on some of the comments, and it seems better now. In the present form, it is clearer the clinical evolution and the discussion of all implications from an athlete's point of view.
- Thank you for valuable input.
Still, authors should correct the number of figure 2 inside the moved text (still, there are references to figure 1 in lines128-135)
- Thank you very much to pick-up the errors, all figure-1 is changed to Figure-2 in the case description.
For keywords, I proposed not to use COVID-19 also in the third word and SARS-CoV-2 in the fourth as already the first two words describe the virus and the disease. That was the reason for my comment.
- Thank you for your valuable suggestion. We changed the key words and removed first COVID-19.
Reviewer 4 Report
Now that you have clarified it, I wonder if the audience should be pediatric cardiologist rather than the audience of this journal. We need to be careful not to send the wrong message. As this patient was symptomatic (chest pain and palpitation) she would have been referred to pediatric cardiology based on AAP/ACC guidance.
I am still not sure how to classify the case. While changes were found on MRI, there is no documented inflammation (all reported labs were normal). What is the differential diagnosis for edema on cardiac MRI. Are there any other possibilities?
As the author knows it is not unusual to have normal echocardiogram in the presence of abnormal MRI. I have several patients with concerning history, abnormal ECG and MRI but normal echocardiogram. The laboratory values were normal when checked as they presented late to the cardiologist.
Please clarify the cycle of the COVID PCR? If it is low cycle then one can conclude that COVID viral load was significantly high at time of presentation. Thank you for providing a copy of the ECG. While I am biased, the T waves are not 100% normal, T waves in leads I (inverted), V5 and V6 (nonspecific changes) are not normal. That being said, I would have brought the patient for repeat ECG in few months rather than obtaining a cardiac MRI. Also with palpitation, why not obtain an event recorder or Holter monitor. Its interesting that you were able to get insurance approval based on this clinical scenario. Kudos to you!
The presentation included "repeat" ECG and echocardiogram, why not include the original echocardiogram. and ECG.
Lastly, this patient meets CDC criteria for MIS-C apart from lack of documented inflammation. The degree of myocardial dysfunction does not explain the 82% pulse ox. The condition was diagnosed 2 months after COVID
Author Response
Thank you for useful comments. I appreciated your valuable time and contribution.
Now that you have clarified it, I wonder if the audience should be pediatric cardiologists rather than this journal's audience. We need to be careful not to send the wrong message. As this patient was symptomatic (chest pain and palpitation), she would have been referred to pediatric cardiology based on AAP/ACC guidance.
- Thank you for your comment. A rheumatologist diagnosed the patient with "post-covid syndrome," her symptoms are chronic, and it was difficult to point out what exactly was going on. The patient was rightly referred to a pediatric cardiologist.
- Regarding the audience of this special issue: Submitted to the section: Global and Public Health,
- https://www.mdpi.com/journal/children/sections/public_global_health
- Cardiac Manifestation in Multisystem Inflammatory Syndrome in Children during
- Global SARS-CoV-2 Pandemic
- https://www.mdpi.com/journal/children/special_issues/Cardiac_Manifestation_Multisystem_Inflammatory_Syndrome_Children_SARS-CoV-2
- I am the editor of this special issue, and the goal is to attract all audiences, including pediatricians, cardiologists, and other allied specialists involved in the care of children with SARS-CoV-2 infection.
I am still not sure how to classify the case. While changes were found on MRI, there is no documented inflammation (all reported labs were normal). What is the differential diagnosis for edema on cardiac MRI. Are there any other possibilities?
- As I stated in the discussion on recent autopsy findings in adults (Linder et al., Ref#10), the myocardial involvement by SARS-CoV-2 is varying from no inflammation to myocyte necrosis. The myocardial disease is not yet well defined in children. Initially, a Kawasaki disease (KD) type, KD-shock syndrome, then MIS-C are described. This is the purpose of this case presentation to highlight different presentations in children.
- Regarding the differential diagnosis of myocardial edema: The lake Louise Criteria as in reference #4: states that edema by T-2 mapping relaxation time suggests acute myocarditis, especially if the ratio of T2 value compared to skeletal muscle is >1.9: we added the value >2.4 (Additional Ref to estimate the ratio of T2 compared to skeletal muscle (adjacent pectoralis muscle): Abdel-Aty H, Boye P, Zagrosek A, Wassmuth R, Kumar A, Messroghli D, Bock P, Dietz R, Friedrich MG, Schulz-Menger J. Diagnostic performance of cardiovascular magnetic resonance in patients with suspected acute myocarditis: comparison of different approaches. J Am Coll Cardiol. 2005;45:1815–22. )
- Edema in the absence of fibrosis suggests reversible injury.
- Edema in the presence of early gadolinium enhancement is indicative of hyperemia and acute myocarditis.
- The differential diagnosis of myocardial edema, but in the context of this case, these are excluded. For example, patients with chronic HIV infection with low-grade viremia can produce patchy myocardial edema. Also, in mitochondrial cardiomyopathies, a predominant increase in extracellular fluid volume is found.
As the author knows, it is not unusual to have a normal echocardiogram in the presence of an abnormal MRI. I have several patients with concerning history, abnormal ECG and MRI but normal echocardiogram. The laboratory values were normal when checked as they presented late to the cardiologist.
- Agree with the reviewer's comments. The MRI was done late, about two months after initial encounter with SARS-CoV-2, one can speculate that the edema may be getting better. Unfortunately, as the medical community is focused on acute presentation, CDC or WHO have not yet defined the criteria for "post-COVID syndrome," less how to approach them. In post-COVID syndrome, many patients, especially common in adults, suffer from fatigue, joint pain, unspecified chest pain, brain fog, and difficulty breathing.
Please clarify the cycle of the COVID PCR? If it is low cycle, then one can conclude that COVID viral load was significantly high at the time of presentation. Thank you for providing a copy of the ECG. While I am biased, the T waves are not 100% normal, T waves in leads I (inverted), V5, and V6 (nonspecific changes) are not normal. That being said, I would have brought the patient for repeat ECG in few months rather than obtaining a cardiac MRI. Also with palpitation, why not obtain an event recorder or Holter monitor. Interestingly, you were able to get insurance approval based on this clinical scenario. Kudos to you!
- Thank you for pointing out an important concept of PCR cycle threshold. I agree with you that COVID patients' cycle threshold values could benefit patients, physicians, and their community. In this case, I have serial Rt-PCR for antigen tests , which are negative after two weeks and at two months. I do not have the patients' cycle threshold values for the virus. I will strongly invite you to write a review article on the "Usefulness of PCR Cycle Threshold in Children," and please submit to this special issue. I believe the readers will greatly appreciate it.
- This patient had a Holter monitor analyzed from a telemetry monitor. ECG and Holter are read by Texas Children’s EP team. They all read as normal. Holter did not show any arrhythmia, but the baseline average heart rate was 92/min.
The presentation included "repeat" ECG and echocardiogram, why not include the original echocardiogram and ECG.
- The original ECG and ECHO were done outside the facility, and I have only reports. I trust those reports and repeat testing were also normal.
Lastly, this patient meets CDC criteria for MIS-C apart from lack of documented inflammation. The degree of myocardial dysfunction does not explain the 82% pulse ox. The condition was diagnosed 2 months after COVID
- This patient does not meet the criteria for MIS-C as there was no fever when she was infected. There has never been an increase in inflammatory markers. The cause of decreased saturation and is not explained. That is the reason the case is presented for publication to highlight the broad latitude of symptoms with COVID-19. This patient had mild abnormal PFT as described in the case description but not reversible by bronchodilator challenge. The patient had normal Chest CT and is being seen by Pulmonology.